# Large-scale statistical dissection of sequence-derived biochemical features distinguishing soluble and insoluble proteins

Nguyen Huy Hoang Vu[1], Bao Long Nguyen[2]*

**1** Faculty of Mathematics, Mechanics and Informatics, University of Science, Vietnam National University, Hanoi (VNU-HUS), Hanoi, Vietnam, **2** School of Economics and Management, Hanoi University of Science and Technology (HUST), Hanoi, Vietnam

* baolongnguyen243@gmail.com

## Abstract

Protein solubility critically influences recombinant expression efficiency and downstream biotechnological applications. While deep learning models have improved predictive accuracy, the intrinsic magnitude, redundancy, and interpretability of classical sequence-derived determinants remain insufficiently characterized. We performed a large-scale univariate analysis on a curated dataset of 78,031 proteins (46,450 soluble; 31,581 insoluble). Thirty-six biochemical descriptors were evaluated using Mann-Whitney U tests with Benjamini-Hochberg false discovery rate correction. Effect sizes were quantified using Cliff's $\delta$, and discriminative performance was assessed by ROC AUC. Although 34 features remained statistically significant after correction, most exhibited small effect sizes and substantial overlap between classes. The strongest effects were associated with size-related features (sequence length and molecular weight; $\delta \approx -0.21$), whereas charge-related descriptors, particularly the proportion of negatively charged residues ($\delta = 0.150$; AUC $= 0.575$), showed consistent but modest shifts. Spearman correlation analysis revealed near-complete redundancy among major size-related variables ($\rho$ up to 0.998). Applying a redundancy threshold ($|\rho| \geq 0.85$), we derived a parsimonious composite integrating sequence length and negative charge proportion, achieving AUC $= 0.624$ (MCC $= 0.1746$). These findings suggest that sequence-level solubility information is consistent with a low-dimensional organization at the level of global sequence-derived descriptors and governed by coordinated weak effects, establishing a transparent statistical baseline for large-scale solubility characterization.

## Introduction

Protein solubility is a fundamental physicochemical property that governs protein folding, intracellular stability, recombinant expression efficiency, and downstream biotechnological applications [1–3]. In heterologous expression systems, poor

**Data availability statement:** All relevant data are within the manuscript and its Supporting information files and URLs.

**Funding:** The author(s) received no specific funding for this work.

**Competing interests:** The authors have declared that no competing interests exist.

solubility frequently results in aggregation, inclusion body formation, reduced functional yield, and substantial experimental cost [4,5]. From a molecular perspective, solubility reflects the balance between intermolecular attractive forces that promote aggregation and intramolecular stabilizing interactions that favor properly folded conformations. Understanding how this balance is encoded in the primary amino acid sequence therefore remains a central problem in computational and structural biology.

A substantial body of experimental and computational work has shown that protein solubility is influenced by multiple sequence-derived properties. Chain length affects folding complexity, translational burden, and the likelihood of partially folded intermediates. Amino acid composition shapes the overall physicochemical landscape, including hydrophobicity, polarity, and charge distribution. Electrostatic interactions contribute to colloidal stability, while hydrophobic clustering and aggregation-prone segments facilitate intermolecular association [6,7]. Importantly, these determinants are not independent: many classical descriptors are structurally or mathematically coupled, suggesting that observed effects may reflect shared latent physicochemical axes rather than distinct mechanisms.

Early computational approaches captured these determinants using interpretable sequence-derived descriptors such as amino acid composition, secondary structure propensity scales, and hydropathy indices [8,9]. These features remain attractive due to their transparency and computational efficiency. However, their relative contribution, redundancy structure, and practical magnitude remain insufficiently characterized at large-scale. In high-powered datasets, extremely small $p$-values can arise from negligible distributional shifts, making it difficult to distinguish statistical significance from biological relevance [10].

Recent advances in machine-learning, particularly deep neural networks and protein language models, have achieved strong predictive performance in solubility benchmarks [11–14]. Despite their accuracy, these approaches often operate as high-capacity black boxes, obscuring the marginal contribution and interaction structure of individual physicochemical features. Moreover, such models typically require substantial computational resources, limiting their applicability in experimental settings where rapid and accessible prediction is needed.

A key unresolved question is whether sequence-level determinants of solubility act primarily through additive contributions of independent physicochemical factors, or whether they depend on higher-order sequence patterns that cannot be captured by classical descriptors. While higher-order representations may improve predictive performance, they introduce additional modeling layers, increase computational complexity, and may accumulate prediction error when inferred from primary sequence alone.

In parallel, many widely accepted determinants of solubility have been identified in relatively small or model-specific datasets, and their importance is often interpreted qualitatively. As a result, their true effect size, redundancy, and practical discriminative contribution remain unclear at scale. This motivates a systematic and statistically rigorous re-evaluation of classical sequence-derived features in a large and heterogeneous dataset.

In the present study, we perform a large-scale statistical dissection of 36 sequence-derived biochemical descriptors using a curated benchmark of soluble and insoluble proteins. Rather than developing a high-capacity predictive model, our objective is to quantify effect magnitude, redundancy structure, and practical discrimination under strict statistical control. By combining non-parametric testing, FDR correction, effect size estimation, ROC analysis, and correlation-based redundancy analysis, we examine whether protein solubility is driven by dominant single determinants or by coordinated weak signals across a low-dimensional physicochemical space.

This study provides a statistically principled and interpretable baseline framework for protein solubility derived from primary sequence features. By integrating large-scale effect size characterization with a reproducible analytical pipeline, it quantifies the practical magnitude and limitations of classical biochemical descriptors. Rather than proposing a higher-capacity predictor, the framework helps characterize the level of discrimination achievable within the restricted space of global sequence-derived physicochemical features, providing a transparent reference for assessing the added value of more complex predictive models.

Importantly, this study is not intended to develop or benchmark a predictive model. Instead, the objective is to quantify the intrinsic statistical structure of sequence-derived physicochemical features under minimal modeling assumptions. Any discriminative performance reported in this work should therefore be interpreted as a descriptive property of the feature space rather than as an estimate of predictive generalization.

## Materials and methods

### Dataset

The dataset was obtained from the curated solubility benchmark introduced by Zhang *et al.* (2024) [14], comprising more than 78,000 protein sequences annotated as soluble or insoluble. The original release provides independent training, validation, and test splits constructed for predictive benchmarking, distributed as three FASTA files containing approximately 74,000, 2,000, and 2,000 protein sequences, respectively.

For the present analysis, all splits were merged into a single dataset because the objective was large-scale distributional characterization rather than predictive generalization. The final dataset comprised 78,031 protein sequences (46,450 soluble and 31,581 insoluble). All sequences and labels are publicly available via Zenodo [14]. Class imbalance in the dataset is unlikely to materially affect rank-based statistics such as the Mann–Whitney U test or ROC AUC.

Protein sequences are represented as contiguous strings of single-letter amino acid codes (e.g., "ACDEFGHIK…") without gaps or alignment. No additional filtering or preprocessing was applied, as the dataset had already been curated in the original benchmark study. In particular, no filtering based on sequence length, redundancy, or sequence identity was performed, and all sequences provided in the original dataset were retained.

Two sequences containing rare or ambiguous amino acid symbols ("X") were preserved. These residues were excluded from amino acid frequency calculations but did not affect other sequence-derived descriptors. All features were computed directly from the raw FASTA sequences.

The FASTA files used in this study correspond exactly to those provided in the original benchmark without modification.

### Feature extraction

All protein sequences were retained in their original form; ambiguous residues were preserved in the raw sequences but excluded from relevant feature calculations as described below. Feature extraction was performed using the 20 canonical amino acids; rare or ambiguous residues (e.g., "X") were not removed from the sequences but were excluded from frequency-based calculations to avoid distortion of compositional features. The presence of such residues was minimal and did not affect global descriptors beyond amino acid frequency measures.

A total of 36 sequence-derived biochemical features were computed for each protein, including:

- **Amino acid frequency features (20 variables).** For each amino acid $a \in \mathcal{A}$ (20 canonical residues), the frequency is defined as:

$$f_a = \frac{n_a}{L},$$

where $n_a$ is the number of occurrences of residue $a$ in the sequence and $L$ is the sequence length (excluding ambiguous residues such as "X"). These variables quantify compositional bias at the residue level.

- **Functional residue group ratios.** Residues were grouped according to physicochemical properties, including: positively charged ($\{K,R,H\}$), negatively charged ($\{D,E\}$), polar ($\{S,T,N,Q\}$), hydrophobic ($\{A,V,I,L,M,F,W,Y\}$), small ($\{A,G,S,T\}$), and sulfur-containing ($\{C,M\}$). For each group $G$, the ratio is defined as:

$$r_G = \frac{\sum_{a \in G} n_a}{L}.$$

These descriptors capture coarse-grained physicochemical composition.

- **Global physicochemical descriptors.** Molecular weight was computed as the sum of residue masses. The isoelectric point (pI) and net charge at pH 7 were estimated using standard residue pKa values and the Henderson–Hasselbalch equation. Mean hydropathy was computed as the average Kyte–Doolittle hydropathy index across the sequence [9]. These variables characterize global physicochemical properties of the sequence.

- **Secondary structure propensity proxies.** Helix, sheet, and turn propensities were approximated by averaging residue-level Chou–Fasman parameters across the sequence [8]. For a given structural class $s$, the feature is:

$$P_s = \frac{1}{L} \sum_{i=1}^{L} p_s(a_i),$$

where $p_s(a_i)$ is the Chou–Fasman propensity of residue $a_i$ for structure $s$.

- **Intrinsic disorder proxy.** Intrinsic disorder tendency was approximated using residue-level disorder propensity scales derived from established literature [15,16]. The disorder ratio was computed as the fraction of residues exceeding a predefined disorder propensity threshold:

$$r_{\text{disorder}} = \frac{1}{L} \sum_{i=1}^{L} \mathbf{1}(d(a_i) > \tau),$$

where $d(a_i)$ denotes the disorder score and $\tau$ is a fixed threshold. The exact propensity scale and threshold used are provided in Table S1 in S1 File and in the accompanying code repository.

- **Aggregation-related proxy.** Aggregation propensity was approximated by the longest contiguous hydrophobic segment in the sequence. Let $\ell_{\max}$ denote the length of the longest run of residues in the hydrophobic set. The feature is defined as:

$$r_{\text{agg}} = \frac{\ell_{\max}}{L}.$$

This proxy captures local clustering of hydrophobic residues associated with aggregation risk.

These descriptors form a structured feature matrix for subsequent statistical evaluation.

## Analytical workflow

The analytical workflow was designed to quantify practical effect magnitude, control false discoveries, eliminate redundancy, and subsequently construct an interpretable composite index.

For each of the 36 sequence-derived descriptors, distributional differences between soluble and insoluble proteins were first evaluated using the Mann-Whitney U test [17]. Resulting $p$-values were adjusted via the Benjamini-Hochberg procedure to control the false discovery rate (FDR) [18]. Effect sizes (Cliff's $\delta$), which measure stochastic dominance without distributional assumptions [19], were computed for all features. For features meeting FDR significance, effect magnitudes were interpreted alongside confidence intervals. Median shifts were expressed using the Hodges–Lehmann estimator with 95% confidence intervals [20]. Stability of $\delta$ was assessed via percentile bootstrap resampling ($B = 2{,}000$) [21]. Univariate discriminative capacity was evaluated using ROC AUC [22], with optimal thresholds determined by Youden's $J$ statistic [23].

Features were subsequently ranked according to absolute $\delta$ to identify descriptors exhibiting the strongest practical separation. Because high-effect features may capture overlapping physicochemical dimensions, pairwise redundancy was assessed prior to composite modeling using Spearman's rank correlation coefficient [24,25]. Spearman correlation was selected due to its robustness to non-normality and its ability to capture monotonic relationships.

Feature pairs exhibiting strong monotonic association ($|\rho| \geq 0.85$) were considered redundant. Within correlated clusters, only a single representative descriptor was retained to avoid double-counting latent physicochemical axes. This redundancy-aware filtering ensured that subsequent integration combined orthogonal dimensions rather than correlated proxies of the same underlying property.

The final composite-$\delta$ index was constructed as a linear combination of retained features, robustly scaled using the median and interquartile range (IQR) [26,27], and weighted by their corresponding $\delta$ values. Proteins were classified according to the sign of the composite score.

The composite-$\delta$ index is not a trained predictive model. Its construction is fully data-dependent and based on empirical effect size estimates derived from the same dataset. Because no parameter fitting or hyperparameter optimization is performed, the resulting index summarizes distributional separability under a statistically controlled and redundancy-filtered feature set.

Consequently, any performance metrics (e.g., AUC, MCC) reported for this index do not represent out-of-sample predictive performance, but rather quantify descriptive separability within the observed feature space.

## Results

### Dataset overview and multiple-testing adjusted significance

An overview of the analytical workflow is provided in Fig 1. Following preprocessing and feature extraction, the final analysis dataset comprised $N = 78{,}031$ protein sequences, including 46,450 soluble and 31,581 insoluble entries. For each sequence, 36 sequence-derived biochemical descriptors were computed, spanning composition, charge-related variables, hydrophobicity measures, structural propensities, and aggregation proxies.

Group-wise distributional differences were evaluated using the Mann-Whitney U test [17], with Benjamini-Hochberg correction to control the false discovery rate (FDR) [18]. After adjustment, 34 of 36 features remained statistically significant ($q < 0.05$), indicating that most classical sequence-level descriptors exhibit detectable distributional shifts between soluble and insoluble proteins.

However, given the large sample size, statistical significance alone is insufficient to infer biological relevance. In high-powered datasets, even negligible shifts can produce extremely small $p$-values [10]. We therefore focus on two

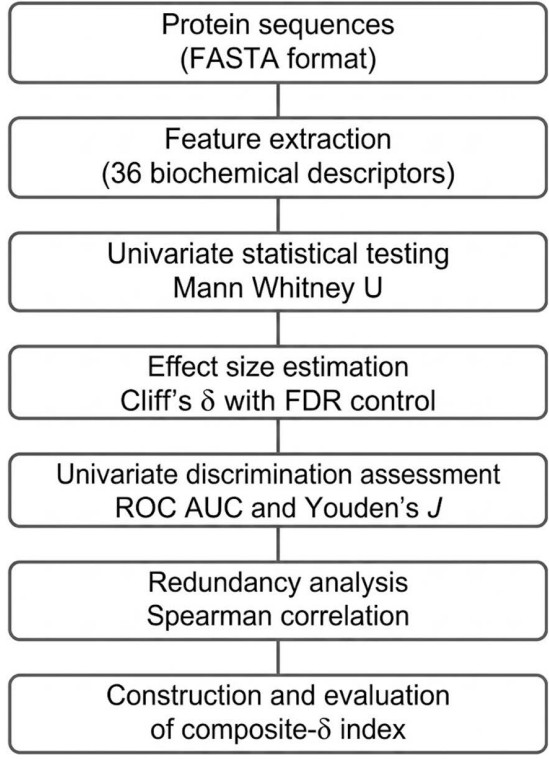

**Fig 1. Analytical workflow.**

complementary quantities: (i) effect magnitude quantified by Cliff's $\delta$ [19], which measures stochastic dominance independent of distributional assumptions, and (ii) univariate discriminative capacity assessed via ROC AUC [22], which provides a descriptive measure of class separability.

To express group differences on the original measurement scale, we report the Hodges–Lehmann (HL) location shift with 95% confidence intervals [20]. Uncertainty of $\delta$ estimates was evaluated using percentile bootstrap intervals ($B = 2,000$) [21]. For completeness, Youden's $J$ statistic and its corresponding optimal threshold $T$ are also reported [23]; in the presence of substantial distributional overlap, $J$ remains close to zero even when statistical significance is achieved.

Collectively, this framework distinguishes statistical detectability from practical magnitude, enabling a biologically grounded interpretation of whether sequence-derived physicochemical features exert strong or merely subtle influence on protein solubility.

## Global physicochemical features

Table 1 summarizes 16 global and grouped-composition descriptors. The largest absolute effects were observed for `length` and `molecular_weight`, both exhibiting $\delta < 0$, indicating that insoluble proteins are, on average, longer and heavier. On the original measurement scale, the Hodges–Lehmann estimates correspond to a median shift of −70 amino acids (95% CI [−77, −62]) for `length` and −6,322 Da (95% CI [−7,148, −5,785]) for `molecular_weight`.

From a mechanistic standpoint, increased chain length elevates folding complexity, prolongs exposure of partially folded intermediates, and increases the probability of intermolecular encounters that nucleate aggregation [1,4]. Larger proteins also present greater solvent-accessible surface area and a higher combinatorial risk of hydrophobic clustering. Despite these biologically plausible trends, univariate discrimination remained limited (AUC = 0.392 and 0.393,

respectively). Values below 0.5 reflect reversed directionality rather than meaningful separability; indeed, inverting the decision rule yields AUC ≈ 0.607, yet substantial distributional overlap persists, as confirmed by small Youden's $J$ statistics.

Charge-related descriptors exhibited coherent and biologically interpretable shifts. The proportion of negatively charged residues (`neg_ratio`) was enriched in soluble proteins ($\delta$ = 0.150, 95% CI [0.134,0.173], AUC = 0.575; HL = 0.00810 with 95% CI [0.00674,0.00937]). Conversely, `isoelectric_point` and `net_charge_pH7` showed $\delta < 0$, indicating higher $pI$ and reduced net negativity among insoluble proteins. These findings align with electrostatic stabilization theory: increased net negative charge enhances intermolecular repulsion, reduces colloidal attraction, and suppresses aggregation propensity [4–6]. The moderate AUC values (≈0.57) indicate measurable but limited standalone discriminatory power.

Hydrophobicity-related descriptors exhibited smaller effects. Both `mean_hydropathy` and `hydrophobic_ratio` showed $\delta < 0$, consistent with the role of exposed hydrophobic patches in promoting solvent exclusion and intermolecular association [7,9]. However, effect sizes were modest and AUC values remained close to 0.5, underscoring extensive overlap between soluble and insoluble distributions. These observations suggest that global hydrophobic averages are insufficient to capture context-dependent aggregation mechanisms.

**Table 1. Results for 16 global sequence-derived features.**

| Feature | Soluble (median [Q1,Q3]) | Insoluble (median [Q1,Q3]) | HL [95% CI] | $\delta$ [95% CI] | FDR $q$ | AUC | $J$ | $T^*$ |
|---|---|---|---|---|---|---|---|---|
| length | 209 [129, 338] | 277 [172, 426] | −70 [−77, −62] | −0.215 [−0.237, −0.196] | 0.000 | 0.392 | 0.158 | 227 |
| molecular_weight | 19 743 [12 201, 31 715] | 25 966 [16 148, 39 869] | −6322.485 [−7147.547, −5784.646] | −0.214 [−0.244, −0.204] | 0.000 | 0.393 | 0.157 | 21520 |
| aggregation_ratio | 0.02262 [0.01502, 0.03371] | 0.01869 [0.01252, 0.02857] | 0.004 [0.003, 0.004] | 0.166 [0.138, 0.178] | 0.000 | 0.583 | 0.125 | 0.020 |
| neg_ratio | 0.1298 [0.1104, 0.1510] | 0.1219 [0.1034, 0.1414] | 0.008 [0.007, 0.009] | 0.150 [0.134, 0.173] | 0.000 | 0.575 | 0.114 | 0.126 |
| isoelectric_point | 6.401 [4.971, 8.357] | 6.931 [5.384, 8.941] | −0.588 [−0.702, −0.468] | −0.132 [−0.167, −0.125] | 0.000 | 0.434 | 0.102 | 6.523 |
| net_charge_pH7 | −1.882 [−7.066, 3.770] | −0.207 [−5.000, 6.476] | −1.618 [−1.931, −1.217] | −0.120 [−0.143, −0.099] | 0.000 | 0.440 | 0.107 | 2.413 |
| mean_hydropathy | −0.3511 [−0.4695, −0.2277] | −0.3085 [−0.4388, −0.1834] | −0.046 [−0.060, −0.035] | −0.080 [−0.106, −0.064] | 0.000 | 0.460 | 0.067 | −0.305 |
| sulfur_ratio | 0.03448 [0.01961, 0.04712] | 0.03681 [0.02193, 0.05013] | −0.002 [−0.003, −0.002] | −0.076 [−0.096, −0.056] | 0.000 | 0.462 | 0.066 | 0.029 |
| helix_prop_mean | 1.029 [1.019, 1.038] | 1.025 [1.015, 1.034] | 0.004 [0.003, 0.005] | 0.060 [0.049, 0.089] | 0.000 | 0.530 | 0.054 | 1.041 |
| hydrophobic_ratio | 0.4019 [0.3786, 0.4253] | 0.4063 [0.3829, 0.4286] | −0.003 [−0.005, −0.001] | −0.051 [−0.053, −0.012] | 0.000 | 0.475 | 0.046 | 0.419 |
| sheet_prop_mean | 0.9937 [0.9857, 1.0024] | 0.9963 [0.9883, 1.0048] | −0.002 [−0.003, −0.000] | −0.050 [−0.056, −0.015] | 0.000 | 0.475 | 0.046 | 1.008 |
| disorder_ratio | 0.3976 [0.3690, 0.4310] | 0.3937 [0.3662, 0.4274] | 0.003 [0.001, 0.005] | 0.048 [0.021, 0.063] | 0.000 | 0.524 | 0.039 | 0.412 |
| polar_ratio | 0.2006 [0.1765, 0.2247] | 0.2031 [0.1798, 0.2278] | −0.002 [−0.005, −0.001] | −0.024 [−0.055, −0.013] | 0.000 | 0.488 | 0.023 | 0.195 |
| tiny_ratio | 0.1389 [0.1111, 0.1667] | 0.1411 [0.1127, 0.1695] | −0.000 [−0.003, 0.002] | −0.021 [−0.035, 0.009] | 0.000 | 0.489 | 0.037 | 0.171 |
| pos_ratio | 0.1412 [0.1212, 0.1636] | 0.1419 [0.1230, 0.1636] | −0.000 [−0.002, 0.001] | −0.015 [−0.026, 0.015] | 0.000 | 0.492 | 0.022 | 0.116 |
| turn_prop_mean | 0.9667 [0.9584, 0.9759] | 0.9672 [0.9589, 0.9763] | −0.001 [−0.003, 0.000] | −0.011 [−0.036, 0.004] | 0.011 | 0.495 | 0.021 | 0.928 |

## Disorder and secondary structure proxies

Intrinsic disorder tendency, approximated by `disorder_ratio`, exhibited a small positive effect ($\delta$ = 0.048, AUC = 0.524), suggesting slight enrichment of disorder-associated residues among soluble proteins. Disordered or flexible regions may increase solvent accessibility and modulate intermolecular interaction landscapes, although their impact appears context-dependent and modest in magnitude [15,16].

Secondary structure propensity proxies (`helix_prop_mean`, `sheet_prop_mean`, and `turn_prop_mean`) demonstrated very small effect sizes. This indicates that simple global aggregation of Chou-Fasman propensities across full-length sequences does not yield strong univariate separation in a large and heterogeneous dataset [8].

## Amino acid composition

Table 2 reports results for the 20 amino acid frequency variables. Many compositional descriptors achieved extremely small *q*-values, reflecting high statistical power; however, most displayed small |$\delta$| and AUC values only marginally deviating from 0.5. Notably, `freq_E` and `freq_D` were elevated in soluble proteins, consistent with the `neg_ratio` signal, whereas `freq_R` and `freq_C` were relatively enriched in insoluble proteins. These trends are compatible with electrostatic contributions and residue-specific side chain chemistry influencing folding stability and aggregation kinetics [4,6,7]. Only two features, `freq_M` and `freq_T`, did not remain significant after FDR correction ($q \geq 0.05$), with $\delta \approx 0$ and AUC $\approx$ 0.5, indicating negligible univariate discriminatory information within this dataset.

## Emergent weak-signal regime and biological interpretation

Taken together, these findings indicate that protein solubility is not governed by a dominant sequence-level determinant but instead reflects coordinated contributions of multiple weak physicochemical signals [2,5]. Size-related variables

**Table 2. Results for amino acid frequency features.**

| Feature | Soluble (median [Q1,Q3]) | Insoluble (median [Q1,Q3]) | HL [95% CI] | $\delta$ [95% CI] | FDR q | AUC | J | T* |
|---------|--------------------------|----------------------------|-------------|-------------------|-------|-----|---|-----|
| freq_C | 0.01017 [0.00312, 0.01882] | 0.01282 [0.00615, 0.02143] | −0.003 [−0.003, −0.002] | −0.131 [−0.151, −0.111] | 0.000 | 0.434 | 0.100 | 0.000 |
| freq_E | 0.07064 [0.05488, 0.08889] | 0.06529 [0.05072, 0.08187] | 0.006 [0.005, 0.007] | 0.121 [0.119, 0.160] | 0.000 | 0.560 | 0.092 | 0.071 |
| freq_R | 0.05350 [0.03663, 0.07273] | 0.05894 [0.04061, 0.07944] | −0.005 [−0.006, −0.005] | −0.105 [−0.101, −0.061] | 0.000 | 0.447 | 0.083 | 0.060 |
| freq_K | 0.06019 [0.04255, 0.07843] | 0.05350 [0.03636, 0.07339] | 0.007 [0.006, 0.008] | 0.091 [0.067, 0.109] | 0.000 | 0.546 | 0.080 | 0.049 |
| freq_D | 0.05721 [0.04098, 0.07391] | 0.05473 [0.04000, 0.07074] | 0.003 [0.002, 0.003] | 0.080 [0.055, 0.097] | 0.000 | 0.540 | 0.067 | 0.061 |
| freq_S | 0.06349 [0.04724, 0.08065] | 0.06630 [0.05000, 0.08333] | −0.003 [−0.003, −0.002] | −0.068 [−0.092, −0.044] | 0.000 | 0.466 | 0.043 | 0.067 |
| freq_P | 0.04244 [0.02899, 0.05755] | 0.04494 [0.03125, 0.06061] | −0.003 [−0.003, −0.002] | −0.064 [−0.088, −0.040] | 0.000 | 0.468 | 0.050 | 0.048 |
| freq_A | 0.07273 [0.05556, 0.09091] | 0.07650 [0.05825, 0.09524] | −0.004 [−0.005, −0.003] | −0.051 [−0.075, −0.026] | 0.000 | 0.474 | 0.052 | 0.092 |
| freq_Q | 0.03768 [0.02500, 0.05172] | 0.03643 [0.02424, 0.05000] | 0.001 [0.001, 0.002] | 0.042 [0.017, 0.066] | 0.000 | 0.521 | 0.042 | 0.030 |
| freq_Y | 0.03061 [0.01923, 0.04211] | 0.02913 [0.01818, 0.04110] | 0.001 [0.001, 0.002] | 0.041 [0.016, 0.066] | 0.000 | 0.521 | 0.041 | 0.009 |
| freq_W | 0.01000 [0.00000, 0.01754] | 0.01031 [0.00000, 0.01852] | −0.000 [−0.001, 0.000] | −0.039 [−0.063, −0.014] | 0.000 | 0.481 | 0.047 | 0.000 |
| freq_I | 0.05128 [0.03604, 0.06818] | 0.05333 [0.03788, 0.07059] | −0.002 [−0.003, −0.001] | −0.039 [−0.063, −0.014] | 0.000 | 0.481 | 0.047 | 0.045 |
| freq_V | 0.06140 [0.04545, 0.07813] | 0.06349 [0.04688, 0.08000] | −0.002 [−0.003, −0.001] | −0.035 [−0.060, −0.010] | 0.000 | 0.482 | 0.043 | 0.077 |
| freq_N | 0.03922 [0.02632, 0.05263] | 0.04000 [0.02740, 0.05405] | −0.001 [−0.002, −0.000] | −0.033 [−0.058, −0.008] | 0.000 | 0.484 | 0.040 | 0.056 |
| freq_G | 0.06195 [0.04598, 0.07937] | 0.06383 [0.04688, 0.08108] | −0.002 [−0.003, −0.001] | −0.028 [−0.053, −0.003] | 0.000 | 0.487 | 0.039 | 0.031 |
| freq_L | 0.09091 [0.07317, 0.10976] | 0.09195 [0.07407, 0.11111] | −0.001 [−0.002, −0.000] | −0.026 [−0.051, −0.001] | 0.000 | 0.487 | 0.038 | 0.079 |
| freq_F | 0.03636 [0.02532, 0.04878] | 0.03704 [0.02632, 0.05000] | −0.001 [−0.001, −0.000] | −0.018 [−0.043, 0.007] | 0.000 | 0.491 | 0.029 | 0.018 |
| freq_H | 0.02381 [0.01389, 0.03448] | 0.02439 [0.01471, 0.03571] | −0.001 [−0.001, 0.000] | −0.016 [−0.041, 0.009] | 0.000 | 0.492 | 0.029 | 0.012 |
| freq_M | 0.02174 [0.01220, 0.03125] | 0.02174 [0.01282, 0.03125] | 0 [−0.001, 0.001] | −0.003 [−0.028, 0.023] | 0.514 | 0.499 | 0.023 | 0.011 |
| freq_T | 0.05263 [0.03704, 0.06897] | 0.05263 [0.03788, 0.06897] | 0 [−0.001, 0.001] | 0.000 [−0.025, 0.025] | 0.989 | 0.500 | 0.022 | 0.072 |

capture a structural burden axis, while charge-related descriptors reflect electrostatic stabilization; hydrophobic and compositional features contribute smaller contextual effects. The generally small Youden's *J* values across individual descriptors confirm that no single feature provides practical threshold-based separation. Rather, the data support a weak-signal, potentially low-dimensional structure in which overlapping physicochemical axes jointly influence solubility. This observation motivates subsequent redundancy-aware integration of selected descriptors into a parsimonious composite index, as detailed below.

Fig 2 illustrates the ROC curves corresponding to the features with the largest absolute effect sizes. The observed asymmetry between negatively and positively charged residues warrants further interpretation. Enrichment of negatively charged residues in soluble proteins is consistent with established biophysical principles, as increased net negative charge enhances electrostatic repulsion and reduces intermolecular association, thereby promoting solubility [28,29].

In contrast, positively charged residues may participate in non-specific interactions with nucleic acids, membranes, or other cellular components, which can facilitate aggregation under certain conditions.

It is important to note that the dataset is derived from recombinant expression in Escherichia coli, and therefore the observed patterns may partially reflect system-specific constraints. The intracellular environment and expression constraints of this system may introduce biases that favor negatively charged, more soluble constructs. Therefore, the observed effect likely reflects a combination of general physicochemical principles and system-specific expression biases. Generalization to other expression systems or native proteomes warrants further investigation.

## Redundancy structure and composite-$\delta$ refinement

**Spearman redundancy analysis.** Prior to finalizing the composite formulation, pairwise redundancy among selected high-effect features was evaluated using Spearman's rank correlation coefficient [24]. The resulting correlation matrix is shown in Fig 3.

Strong monotonic associations were observed among size-related descriptors. In particular, sequence length and molecular weight exhibited near-complete collinearity ($\rho \approx 0.998$), reflecting their deterministic structural coupling. Aggregation-related metrics also displayed strong monotonic association with size-related variables ($|\rho| \approx 0.9$), indicating that these descriptors largely capture a shared latent structural axis.

In contrast, the proportion of negatively charged residues (`neg_ratio`) showed minimal correlation with size-related variables ($|\rho| < 0.05$), suggesting relative independence of electrostatic and size dimensions. Applying a predefined redundancy criterion ($|\rho| \geq 0.85$), correlated size-related descriptors were considered redundant. To avoid double-counting a single latent physicochemical axis, only one representative size descriptor was retained. We selected `length` due to its direct structural interpretability and deterministic relation to molecular mass. The threshold ($|\rho| \geq 0.85$) was adopted as a conservative criterion to remove strongly collinear features while retaining distinct physicochemical dimensions. A formal dimensionality analysis (e.g., PCA) was not performed and is left for future work.

To illustrate the extent of distributional overlap, Fig 4 presents the density distributions of the two principal orthogonal features: sequence length and negative charge proportion. In both cases, soluble and insoluble proteins exhibit substantial overlap despite statistically significant differences in central tendency.

For sequence length, insoluble proteins display a right-shifted distribution, reflecting a tendency toward longer chains. However, a considerable fraction of soluble proteins occupies the same range, limiting practical separability. Similarly, for negative charge proportion, soluble proteins show a modest enrichment in negatively charged residues, yet the two classes remain strongly intermixed.

These observations provide a direct visual confirmation that, although detectable at scale, the underlying physicochemical differences are small relative to within class variability. This substantial overlap explains the limited univariate discriminative performance observed in ROC analysis and reinforces the interpretation that protein solubility is governed by coordinated weak effects rather than strongly separable individual features.

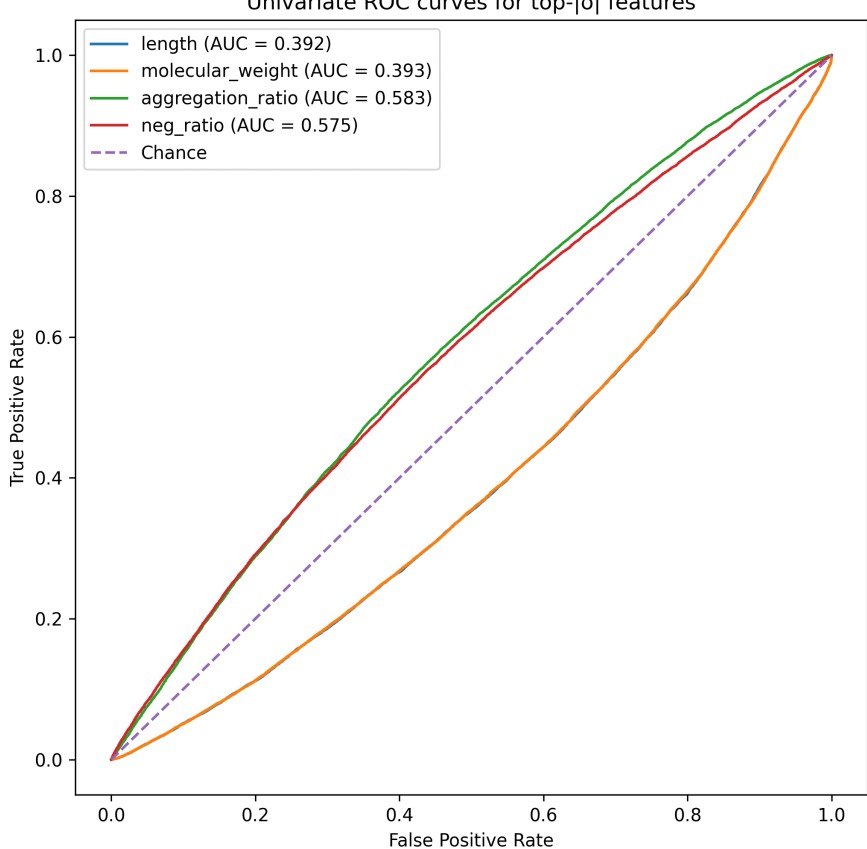

**Fig 2. Univariate ROC curves for the four features with the largest absolute Cliff's $\delta$.**

The initial composite-$\delta$ index integrated the top ranked descriptors by absolute effect size, robustly scaled using median and interquartile range (IQR) to mitigate the influence of outliers.

$$x_j^{(\text{rs})} = \frac{x_j - \text{median}(x_j)}{\text{IQR}(x_j)}, \qquad \text{IQR}(x_j) = Q_{0.75}(x_j) - Q_{0.25}(x_j)$$

(1)

$$S = \sum_{j=1}^{k} \delta_j\, x_j^{(\text{rs})}$$

(2)

$$\hat{y} = \begin{cases} 1, & \text{if } S > 0 \quad \text{(soluble)} \\ 0, & \text{if } S \leq 0 \quad \text{(insoluble)} \end{cases}$$

(3)

**Reduced composite formulation.** Using global medians and IQR values (Table 3), the reduced composite-$\delta$ score becomes:

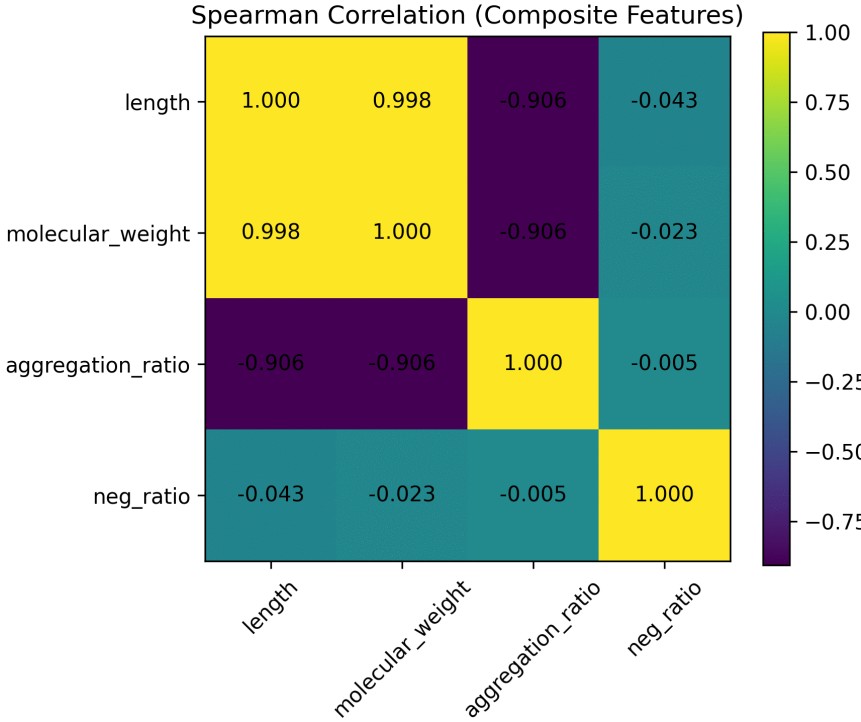

**Fig 3. Spearman correlation matrix among top ranked global features.**

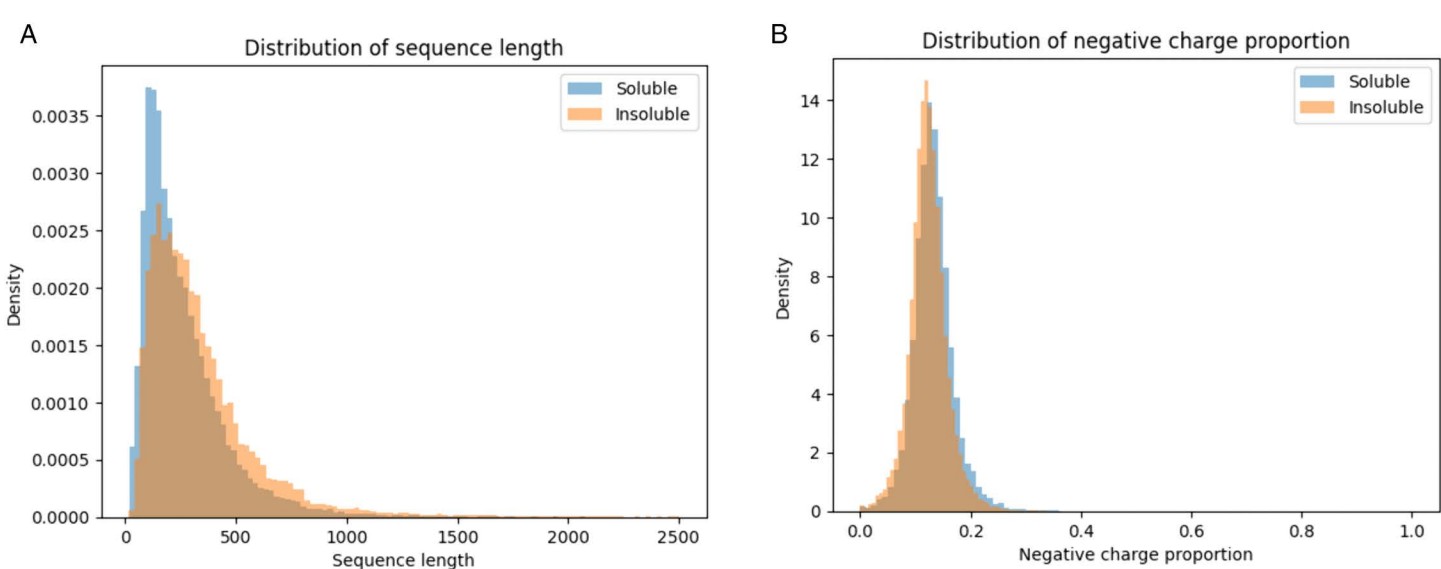

**Fig 4. Distributions of sequence length (left) and negative charge proportion (right) for soluble and insoluble proteins, showing substantial overlap between classes.**

**Table 3. Robust summaries of the two retained physicochemical dimensions defining the reduced composite index.**

| Feature | Median [IQR] |
| --- | --- |
| length | 236 [231] |
| neg_ratio | 0.126531 [0.039934] |

$$S = -0.215 \frac{L - 236}{231} + 0.150 \frac{N_- - 0.126531}{0.039934}$$

(4)

This two-dimensional formulation preserves the dominant size and electrostatic axes while eliminating redundant contributions. As shown below, discriminative performance remains comparable to the full multi-feature composite, supporting the interpretation that sequence-level solubility information is consistent with a low-dimensional organization at the level of global sequence-derived descriptors. The use of a linear formulation is intentional, as the objective is to isolate the contribution of first-order physicochemical features without introducing higher-order interactions that may obscure interpretability.

Performance metrics (AUC and MCC) of reference models were taken directly from the original benchmark study [14] to provide contextual reference. These values are shown alongside the descriptive metrics of the composite-$\delta$ index but do not constitute a direct head-to-head comparison.

The comparison in Table 4 is provided for contextual reference only. The reported performance of the composite-$\delta$ index is not directly comparable to supervised models evaluated under strict train/test separation, as the present analysis does not involve independent validation. Instead, these values should be interpreted as descriptive indicators of signal strength in sequence-derived features.

Because the present study merges the original training, validation, and test splits for distributional analysis rather than predictive evaluation, the reported metrics should not be interpreted as estimates of out-of-sample performance.

Importantly, the proposed composite-$\delta$ index does not constitute a trained predictive model. Its construction is fully data-dependent and based on empirical effect size estimates derived from the same dataset. Consequently, AUC and MCC are reported solely as descriptive measures of separability within the observed feature space, rather than as indicators of predictive generalization.

The purpose of the composite-$\delta$ formulation is therefore not to achieve state-of-the-art predictive performance, but to provide an interpretable statistical reference that characterizes the magnitude and structure of sequence-derived solubility signals.

**Table 4. Contextual performance reference for existing solubility prediction methods and the composite-$\delta$ index.**

| Model | T | AUC | Acc. | F1 | MCC | Prec. | Sens. | Spec. |
| --- | --- | --- | --- | --- | --- | --- | --- | --- |
| Protein_sol | 0.5 | 0.5985 | 0.5649 | 0.6350 | 0.1405 | 0.5470 | 0.7569 | 0.3729 |
| SKADE | 0.5 | 0.6882 | 0.6443 | 0.5304 | 0.3305 | 0.7811 | 0.4015 | 0.8874 |
| SWI | 0.5 | 0.5597 | 0.5423 | 0.6334 | 0.0971 | 0.5284 | 0.7905 | 0.2938 |
| Solupro | 0.5 | 0.7126 | 0.6308 | 0.6639 | 0.2667 | 0.6095 | 0.7290 | 0.5325 |
| EPSOL | 0.5 | 0.6664 | 0.6663 | 0.5915 | 0.3578 | 0.7630 | 0.4830 | 0.8499 |
| NetSolP | 0.5 | 0.6183 | 0.5502 | 0.6557 | 0.1268 | 0.5312 | 0.8564 | 0.2437 |
| DeepSol-E | 0.4 | 0.6660 | 0.5990 | 0.6407 | 0.2036 | 0.5804 | 0.7150 | 0.4830 |
| PLM_Sol | 0.5 | 0.8342 | 0.7299 | 0.7542 | 0.4690 | 0.6919 | 0.8289 | 0.6308 |
| **Composite-$\delta$** | **0** | **0.6240** | **0.5734** | **0.5875** | **0.1746** | **0.6922** | **0.5102** | **0.6663** |

To contextualize computational efficiency, Table 5 provides a qualitative comparison of inference-time complexity across representative models. For clarity, we define the notation used in the complexity analysis as follows: $L$ denotes the sequence length, $d$ the number of input features, $t$ the number of trees or ensemble components, $k$ the kernel or filter size in convolutional models, $H$ the number of transformer layers, $h$ the number of attention heads per layer. All complexity estimates refer to inference-time computational cost under standard implementations.

The redundancy-aware composite-$\delta$ baseline achieved AUC = 0.6240 and MCC = 0.1746. Although this performance remains below that of high-capacity protein language model architectures such as PLM_Sol [14], it is comparable to, and in several cases exceeds, traditional physicochemical feature-based predictors reported in the literature. Importantly, the composite-$\delta$ formulation involves no parameter fitting, no embedding extraction, and no hyperparameter optimization. The decision rule is fully determined by robust scaling [26,27] and statistically estimated effect sizes derived directly from empirical distributions. Consequently, the observed discrimination reflects separability of sequence-level biochemical features rather than model-driven representation learning.

From a computational standpoint, it is important to clarify that the computational complexity of the proposed composite-$\delta$ model depends on the level of representation considered. When applied directly to raw amino acid sequences, feature extraction requires a single pass over the sequence, resulting in a linear time complexity of $O(L)$, where $L$ is the sequence length.

However, once sequence-derived features are computed, the composite-$\delta$ score itself is obtained through a simple linear combination, which operates in constant time $O(1)$. Therefore, the model is best interpreted as having $O(L)$ end-to-end complexity with an $O(1)$ scoring step.

This distinction is important in practice, as feature extraction is computationally inexpensive and scales linearly with sequence length, while the absence of training, parameter optimization, or iterative inference makes the overall framework substantially more efficient than high-capacity models such as protein language models, which typically scale at least quadratically with sequence length. In practical settings, where sequence-derived features are often precomputed or cached, the effective runtime of the scoring step becomes negligible. In contrast, classical machine-learning predictors based on handcrafted descriptors scale at least linearly with the number of features or ensemble components ($O(d)$–$O(t)$) [30]. Convolutional neural networks introduce sequence-length-dependent cost ($O(L \cdot k)$) [31], while transformer-based protein language models incur quadratic complexity with respect to sequence length due to the self-attention mechanism ($O(H \cdot L^2 \cdot h)$) [32]. This progression corresponds to substantially higher computational cost.

The moderate performance gap between composite-$\delta$ and PLM-based architectures therefore reflects a clear trade-off between representational capacity and computational efficiency. While transformer models capture higher-order contextual interactions at substantial resource cost, the present results demonstrate that a low-dimensional,

**Table 5. Qualitative comparison of inference-time computational requirements across representative solubility prediction approaches.**

| Model | Architecture Type | Inference Complexity | Training Requirement |
|---|---|---|---|
| Protein_sol | Feature-based ML | $O(d)$ to $O(t)$ | Supervised training |
| SKADE | Ensemble model | $O(t)$ | Supervised training |
| SWI | Linear sequence index | $O(L)$ | None |
| Solupro | Feature-based ML | $O(d)$ to $O(t)$ | Supervised training |
| EPSOL | Ensemble / ML | $O(t)$ | Supervised training |
| NetSolP | Deep neural network | $O(L \cdot k)$ | Supervised training |
| DeepSol-E | CNN-based model | $O(L \cdot k)$ | Supervised training |
| PLM_Sol | Transformer (PLM-based) | $O(H \cdot L^2 \cdot h)$ | Pretrained + fine-tuned |
| **Composite-$\delta$** | Interpretable linear score | **O(L)(featureextraction) + O(1)(scoring)** | None |

redundancy-controlled linear formulation retains a non-trivial portion of discriminative signal with negligible computational overhead.

From a mechanistic perspective, the achieved performance indicates that global physicochemical descriptors encode a measurable but limited solubility signal. The modest AUC and MCC values are consistent with the substantial distributional overlap observed in univariate analyses and reinforce the interpretation of solubility as a weak-signal, low-dimensional, and context-dependent phenotype [2,5]. Rather than suggesting inadequacy of classical descriptors, these findings clarify their role: global sequence-derived features provide an interpretable empirical reference for the level of discrimination and reflect the interpretable physicochemical axes upon which more complex models may implicitly build. In this sense, the composite-$\delta$ baseline serves as both a transparent reference model and a mechanistic anchor for evaluating the added value of high-capacity predictive frameworks under explicit computational constraints. In addition to these mechanistic insights, it is important to consider the robustness of the proposed framework. To assess robustness, we note that the composite-$\delta$ formulation is derived directly from global distributional properties and does not involve optimization or parameter tuning beyond direct empirical estimation from the same dataset, the framework may be less sensitive to dataset partitioning, although this was not formally evaluated compared to trained models; formal stability assessment is left for future work. Future work may evaluate stability under resampling; however, given the large sample size ($N > 78,000$), variance in estimated effect sizes is expected to be limited, although not formally evaluated.

## Conclusion

This study presents a large-scale, statistically controlled, and interpretable analysis of sequence-derived biochemical determinants of protein solubility using 78,031 labeled proteins. By combining non-parametric testing, multiple-testing correction, effect-size estimation, uncertainty quantification, ROC-based evaluation, and redundancy analysis, we distinguish statistical detectability from practical discriminative relevance.

The results show that soluble and insoluble proteins are statistically distinguishable across many sequence-derived descriptors, but the corresponding effects are generally small and strongly overlapping. Protein solubility at the sequence level therefore appears to lie in a weak-signal regime, where no individual descriptor provides strong standalone discrimination. Instead, separability emerges from the coordinated contribution of multiple physicochemical features with modest individual effects.

This conclusion refines the interpretation of classical solubility determinants such as sequence length, molecular weight, charge, and hydrophobicity. Although these factors are statistically significant, their practical effect sizes are considerably smaller than may be inferred from significance alone. Redundancy analysis further indicates that much of the signal is organized along a limited number of latent physicochemical axes. Size-related descriptors largely reflect a shared structural-burden dimension, whereas charge-related features form a comparatively independent electrostatic axis. After redundancy filtering, a parsimonious two-dimensional composite based on sequence length and negative charge proportion retained measurable descriptive discrimination.

The observed patterns are consistent with predominantly additive contributions of global physicochemical features, although higher-order dependencies were not explicitly modeled. The present framework deliberately relies on global sequence-derived descriptors and therefore does not capture positional or contextual effects. In proteins, the contribution of a residue depends strongly on its local and structural environment; identical amino acids may affect folding stability, aggregation propensity, or solvent exposure differently depending on context [33]. While protein language models can implicitly capture such ordered and higher-order dependencies, they introduce greater computational cost and multi-stage inference. In contrast, this study isolates the intrinsic contribution of first-order sequence descriptors, providing a transparent and computationally efficient lower-bound characterization of sequence-based solubility information.

All dataset partitions were merged to maximize statistical power for descriptive distributional analysis; consequently, no out-of-sample predictive claims are made. The composite-$\delta$ index should therefore be interpreted as a data-dependent

statistical reference rather than a trained predictive model. Although its descriptive discrimination is moderate relative to high-capacity models, it offers a favorable trade-off between interpretability, computational efficiency, and practical applicability. It operates in linear time with respect to sequence length, requires no training, and supports rapid pre-screening in resource-constrained experimental settings.

Overall, this work establishes a statistically principled and interpretable reference framework for sequence-based solubility analysis. By quantifying effect size, redundancy, and low-dimensional structure in a large-scale dataset, it clarifies the role of classical descriptors, refines their biological interpretation, and provides a transparent baseline for assessing the added value of more complex machine-learning and protein-language-model approaches.

## Supporting information

**S1 File. Results section data.**
(ZIP)

**S2 File. Dataset used for the study.**
(CSV)

## Author contributions

**Conceptualization:** Nguyen Huy Hoang Vu, Long Nguyen Bao.

**Data curation:** Nguyen Huy Hoang Vu.

**Formal analysis:** Nguyen Huy Hoang Vu.

**Investigation:** Nguyen Huy Hoang Vu.

**Methodology:** Nguyen Huy Hoang Vu.

**Project administration:** Long Nguyen Bao.

**Resources:** Nguyen Huy Hoang Vu, Long Nguyen Bao.

**Software:** Nguyen Huy Hoang Vu.

**Supervision:** Long Nguyen Bao.

**Validation:** Nguyen Huy Hoang Vu, Long Nguyen Bao.

**Visualization:** Nguyen Huy Hoang Vu.

**Writing – original draft:** Nguyen Huy Hoang Vu.

**Writing – review & editing:** Long Nguyen Bao.

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
