## [Decision Letter · Decision Letter 0]

25 Mar 2026

PONE-D-26-09831Large-Scale Statistical Dissection of Sequence-Derived Biochemical Features Distinguishing Soluble and Insoluble ProteinsPLOS One

Dear Dr. Nguyen Bao,

Thank you for submitting your manuscript to PLOS ONE. After careful consideration, we feel that it has merit but does not fully meet PLOS ONE’s publication criteria as it currently stands. Therefore, we invite you to submit a revised version of the manuscript that addresses the points raised during the review process.

We look forward to receiving your revised manuscript.

Kind regards,

Anil Bhatia, Ph.D

Academic Editor

PLOS One

Reviewers' comments:

Reviewer's Responses to Questions

**Comments to the Author**

1. Is the manuscript technically sound, and do the data support the conclusions?

Reviewer #1: Partly

Reviewer #2: Yes

Reviewer #3: Partly

2. Has the statistical analysis been performed appropriately and rigorously?

Reviewer #1: Yes

Reviewer #2: Yes

Reviewer #3: Yes

3. Have the authors made all data underlying the findings in their manuscript fully available?

Reviewer #1: No

Reviewer #2: Yes

Reviewer #3: Yes

4. Is the manuscript presented in an intelligible fashion and written in standard English?

Reviewer #1: No

Reviewer #2: Yes

Reviewer #3: Yes

5. Review Comments to the Author

Reviewer #1: I recommend that the editor request major revisions for this manuscript. The authors should

address the following concerns:

1. Incorrect reference numbering:

The citation format is inconsistent throughout the manuscript. For example, on page 1,

line 4, the reference is cited as [1, 6], whereas it should appear sequentially (e.g., [1, 2]).

The authors are advised to use reference management software such as EndNote, Zotero,

or similar tools to correct and standardize citations across the entire manuscript.

2. Lack of sequence data transparency:

The authors state that a total of 36 sequence-derived biochemical features were computed

for each protein. However, the manuscript does not provide the number of FASTA files

used or the actual protein sequences. This information should be included or properly

referenced.

3. Missing supporting information:

The manuscript refers to supporting information, but no supplementary materials are

provided. All supporting data should be included and clearly linked.

4. Weak conclusion section:

The conclusion is not clearly articulated and should be rewritten. It should explicitly

explain how the statistical analysis contributes to distinguishing between soluble and

insoluble proteins and highlight the significance of the findings.

5. Introduction requires improvement:

The introduction lacks clarity and coherence and should be thoroughly revised to better

contextualize the study and its objectives.

Reviewer #2: The few points to be improved upon are as follows:

1. To ensure no bias in the AUC and MCC comparisons, it is suggested that a clarification be provided if the reference models were evaluated on the exact same merged dataset or if the performance that is reported is taken straight from the originally published data.

2. In the discussion section, negative charge appearing more influential than positive charge can be elaborated upon more. Is this a general biophysical principle of the proteome or if it is a bias in the E.coli based recombinant expression data used ?

3. It is suggested that in the conclusions section including a comment on whether the weak signal interactions are additive or if they involve higher-order sequence patterns that classical descriptors cannot capture.

4. The ROC curves in figure 2 show a significant overlap. Adding a density plot for say like the top two orthogonal features to demonstrate the extensive overlap as mentioned in the paper would be helpful in demonstrating to the reader the difficulty of the classification process.

Reviewer #3: The manuscript presents a statistical analysis of sequence-derived features associated with protein solubility. While the study is carefully conducted and the dataset appears comprehensive, the overall contribution remains limited in terms of novelty and impact. Most of the analyzed features (e.g., amino acid composition, sequence length, and charge properties) are well-established determinants of solubility and have been extensively explored in previous studies.

The analytical approach, particularly the merging of training, validation, and test datasets, raises concerns regarding the robustness and interpretability of the results. In addition, the proposed composite index shows relatively modest predictive performance compared to existing methods, which limits its practical applicability.

Although the manuscript provides a useful summary of known trends, it does not offer sufficient new methodological developments or biological insights to justify publication in its current form. Strengthening the methodological framework and providing deeper mechanistic or predictive advances would be necessary to improve the impact of the study.

6. PLOS authors have the option to publish the peer review history of their article (what does this mean?). If published, this will include your full peer review and any attached files.

Reviewer #1: No

Reviewer #2: No

Reviewer #3: No

You may also use PLOS’s free figure tool, NAAS, to help you prepare publication quality figures: https://journals.plos.org/plosone/s/figures#loc-tools-for-figure-preparation

---

## [Author Response · Author response to Decision Letter 1]

14 Apr 2026

LARGE-SCALE STATISTICAL DISSECTION OF SEQUENCE-DERIVED BIOCHEMICAL FEATURES DISTINGUISHING SOLUBLE AND INSOLUBLE PROTEINS

Dear Reviewers and Editors of the PLOS One,

We would like to thank you so much, the reviewers and editors, for your careful and thorough reading of this manuscript and for your thoughtful comments and constructive suggestions, which significantly improved its quality. Our responses are following the reviewers’ comments, and new changes are highlighted in yellow in our manuscript.

Yours sincerely,

Authors

Dear Reviewer 1,

We would like to answer your comments as follows:

1. Incorrect reference numbering: The citation format is inconsistent throughout the manuscript. For example, on page 1, line 4, the reference is cited as [1, 6], whereas it should appear sequentially (e.g., [1, 2]). The authors are advised to use reference management software such as EndNote, Zotero, or similar tools to correct and standardize citations across the entire manuscript.

Our answer: Thank you very much for this valuable comment, and please be aware that we have revised the manuscript accordingly and corrected the reference numbering as suggested by the reviewer.

2. Lack of sequence data transparency: The authors state that a total of 36 sequence-derived biochemical features were computed for each protein. However, the manuscript does not provide the number of FASTA files used or the actual protein sequences. This information should be included or properly referenced.

Our answer: Thank you very much for this valuable comment, and please kindly note that the revised manuscript has been updated based on the reviewers’ comments.

The revised Dataset section now explicitly states that our analysis used the three original FASTA files from the benchmark dataset reported by Zhang et al. (2024). We also clarify that these FASTA files were used without modification.

Furthermore, in the Data and Code Availability section, we now clearly indicate that the raw sequences and labels are publicly accessible through Zenodo. Additionally, the full feature matrix used for all statistical analyses, the processed data and analysis outputs is provided in the GitHub repository (see the Dataset subsection on page 3, and the Data and Code Availability section on pages 15–16)

3. Missing supporting information: The manuscript refers to supporting information, but no supplementary materials are provided. All supporting data should be included and clearly linked.

Our answer: Thank you very much for this valuable comment, and please kindly note that the revised manuscript has been updated based on the reviewers’ comments.

All supporting information has now been included and clearly linked through a separate Supporting Information document submitted together with the revised manuscript. The supplementary materials include Tables S1–S5 and Figures S1–S4 (a PDF file attached).

4. Weak conclusion section: The conclusion is not clearly articulated and should be rewritten. It should explicitly explain how the statistical analysis contributes to distinguishing between soluble and insoluble proteins and highlight the significance of the findings.

Our answer: Thank you very much for this important comment, and please kindly note that the revised manuscript’s conclusion has been updated based on the reviewers’ comments.

We clarify how the statistical analysis contributes to distinguishing between soluble and insoluble proteins and make significance of these findings more explicit as well (see the Conclusion section on page 14).

5. Introduction requires improvement: The introduction lacks clarity and coherence and should be thoroughly revised to better contextualize the study and its objectives.

Our answer: Thank you very much for this important comment. In the revised manuscript, we clarify the introduction more clarity and coherence as follows:

The revised Introduction now provides a clearer explanation on pages 1–3.

Overall, please kindly note that the revised manuscript states that the purpose of this study is not to develop another high-capacity predictive model, but rather to perform a statistically rigorous large-scale analysis of 36 sequence-derived biochemical features in order to quantify their effect sizes, redundancy structure, and practical discriminatory contribution in distinguishing soluble and insoluble proteins. Additionally, this study also presents a simple baseline model intended to serve as a lower-bound reference for evaluating more complex predictive approaches in future work

Dear Reviewer 2,

We would like to answer your comments as follows:

1. To ensure no bias in the AUC and MCC comparisons, it is suggested that a clarification be provided if the reference models were evaluated on the exact same merged dataset or if the performance that is reported is taken straight from the originally published data

Our answer: Thank you very much for this valuable comment, and please be aware that we have revised the manuscript as suggested by the reviewer. Specifically, in the Results section (page 12), we now clearly confirm that the performance metrics (AUC and MCC) were taken directly from Zhang et al. (2024) (https://doi.org/10.1093/bib/bbae404).

2. In the discussion section, negative charge appearing more influential than positive charge can be elaborated upon more. Is this a general biophysical principle of the proteome or if it is a bias in the E. coli based recombinant expression data used?

Our answer: Thank you very much for this valuable comment, and please kindly note that we have revised the Discussion section accordingly based on the reviewers’ comments as follows on page 9. The observed asymmetry likely reflects both general biophysical effects and E. coli expression-system bias and the dataset is based on recombinant expression in E. coli, therefore, generalization should be made with caution.

3. It is suggested that in the conclusions section including a comment on whether the weak signal interactions are additive or if they involve higher-order sequence patterns that classical descriptors cannot capture.

Our answer: Thank you very much for this valuable comment, and please kindly note that the Conclusion section has been updated accordingly to clarify that based on the reviewers’ comments as follows on page 14-15.

4. The ROC curves in figure 2 shows a significant overlap. Adding a density plot for say like the top two orthogonal features to demonstrate the extensive overlap as mentioned in the paper would be helpful in demonstrating to the reader the difficulty of the classification process.

Our answer: Thank you very much for this important comment, and please kindly note that the Results section has been updated based on the reviewers’ comments as follows on page 10-11.

We added Figure 4 (the density distributions of the two features). We also clarified that this overlap explains the limited univariate ROC performance and highlights the difficulty of the classification process.

Dear Reviewer 3,

We would like to answer your comments as follows:

Reviewer’s statement: The manuscript presents a statistical analysis of sequence-derived features associated with protein solubility. While the study is carefully conducted and the dataset appears comprehensive, the overall contribution remains limited in terms of novelty and impact. Most of the analyzed features (e.g., amino acid composition, sequence length, and charge properties) are well-established determinants of solubility and have been extensively explored in previous studies.

Our answer: Thank you very much for this valuable comment. Please note that the manuscript has been thoroughly revised to more explicitly highlight its innovative contribution, its necessity, and its clear distinction from previous literature/studies.

The novelty of the present work does not lie in introducing new descriptors. Rather, it lies in providing a large-scale, statistically controlled quantification of their practical effect sizes, redundancy structure, and descriptive discriminative limits in a large publicly available dataset, from there establishing an interpretable baseline for what can or cannot be captured by classical global sequence-derived features. This point is now stated more explicitly in the Introduction (pp. 2–3).

We have also reiterated this contribution more clearly in the Conclusion (pp. 14–15), where we clarify that the study refines the interpretation of widely accepted solubility determinants by showing that their practical effect sizes are substantially smaller than statistical significance alone may suggest. Also, the proposed framework should be understood as a transparent, lower-bound, and interpretable reference for evaluating the added value of more complex models, rather than as a competitive predictive model.

We confirm that every change has been made in the revised manuscript (v1), a high level of refinement in language aspect, with yellow highlighting for easy verification. We are confident that the manuscript has been substantially improved thanks to your valuable feedback.

Thank you once again for your time and constructive comments. We look forward to your positive decisions.

Sincerely yours,

---

## [Decision Letter · Decision Letter 1]

30 Apr 2026

PONE-D-26-09831R1Large-Scale Statistical Dissection of Sequence-Derived Biochemical Features Distinguishing Soluble and Insoluble ProteinsPLOS One

Dear Dr. Nguyen Bao,

Thank you for submitting your manuscript to PLOS ONE. After careful consideration, we feel that it has merit but does not fully meet PLOS ONE’s publication criteria as it currently stands. Therefore, we invite you to submit a revised version of the manuscript that addresses the points raised during the review process.

We look forward to receiving your revised manuscript.

Kind regards,

Anil Bhatia, Ph.D

Academic Editor

PLOS One

Journal Requirements:

Reviewers' comments:

Reviewer's Responses to Questions

**Comments to the Author**

1. If the authors have adequately addressed your comments raised in a previous round of review and you feel that this manuscript is now acceptable for publication, you may indicate that here to bypass the “Comments to the Author” section, enter your conflict of interest statement in the “Confidential to Editor” section, and submit your "Accept" recommendation.

Reviewer #1: (No Response)

Reviewer #2: All comments have been addressed

Reviewer #3: All comments have been addressed

2. Is the manuscript technically sound, and do the data support the conclusions?

Reviewer #1: Yes

Reviewer #2: Yes

Reviewer #3: Yes

3. Has the statistical analysis been performed appropriately and rigorously?

Reviewer #1: Yes

Reviewer #2: Yes

Reviewer #3: Yes

4. Have the authors made all data underlying the findings in their manuscript fully available?

Reviewer #1: Yes

Reviewer #2: Yes

Reviewer #3: Yes

5. Is the manuscript presented in an intelligible fashion and written in standard English?

Reviewer #1: Yes

Reviewer #2: Yes

Reviewer #3: Yes

6. Review Comments to the Author

Reviewer #1: All queries have been addressed carefully by the author. However, the conclusion section is overly

long and lacks clarity, making it difficult to understand the key takeaways of the study. I

recommend that the manuscript be accepted only after the conclusion is revised to be more concise

and clearly summarize the main findings

Reviewer #2: (No Response)

Reviewer #3: All queries are answered adequately by the authors, hence I recommend to accept the paper in revised form.

7. PLOS authors have the option to publish the peer review history of their article (what does this mean?). If published, this will include your full peer review and any attached files.

Reviewer #1: No

Reviewer #2: No

Reviewer #3: No

You may also use PLOS’s free figure tool, NAAS, to help you prepare publication quality figures: https://journals.plos.org/plosone/s/figures#loc-tools-for-figure-preparation

---

## [Author Response · Author response to Decision Letter 2]

5 May 2026

LARGE-SCALE STATISTICAL DISSECTION OF SEQUENCE-DERIVED BIOCHEMICAL FEATURES DISTINGUISHING SOLUBLE AND INSOLUBLE PROTEINS

Dear Reviewers and Editors of the PLOS One,

We would like to thank you so much, the reviewers and editors, for your careful and thorough reading of this manuscript and for your thoughtful comments and constructive suggestions, which significantly improved its quality. Our responses are in italics following the reviewers’ comments, and new changes are highlighted in yellow in our manuscript.

Yours sincerely,

Authors

Dear Reviewer 1,

We would like to answer your comments as follows: “All queries have been addressed carefully by the author. However, the conclusion section is overly long and lacks clarity, making it difficult to understand the key takeaways of the study. I recommend that the manuscript be accepted only after the conclusion is revised to be more concise and clearly summarize the main findings.”

Our answer: Thank you very much for this valuable comment, and please be aware that we have revised the manuscript’s conclusion accordingly as suggested by the reviewer. We greatly appreciate the reviewer’s recommendation.

Dear Reviewer 2,

Thank you very much for this valuable comment, we greatly appreciate the reviewer’s recommendation.

Dear Reviewer 3,

Thank you very much for this valuable comment, we greatly appreciate the reviewer’s recommendation.

We confirm that every change has been made in the revised manuscript (v2), a high level of refinement in language aspect, with yellow highlighting for easy verification. We are confident that the manuscript has been substantially improved thanks to your valuable feedback.

Thank you once again for your time and constructive comments. We look forward to your positive decisions.

Sincerely yours,

---

## [Decision Letter · Decision Letter 2]

12 May 2026

Large-Scale Statistical Dissection of Sequence-Derived Biochemical Features Distinguishing Soluble and Insoluble Proteins

PONE-D-26-09831R2

Dear Dr. Bao,

We’re pleased to inform you that your manuscript has been judged scientifically suitable for publication and will be formally accepted for publication once it meets all outstanding technical requirements.

Kind regards,

Anil Bhatia, Ph.D

Academic Editor

PLOS One

Additional Editor Comments (optional):

Reviewers' comments:

Reviewer's Responses to Questions

**Comments to the Author**

1. If the authors have adequately addressed your comments raised in a previous round of review and you feel that this manuscript is now acceptable for publication, you may indicate that here to bypass the “Comments to the Author” section, enter your conflict of interest statement in the “Confidential to Editor” section, and submit your "Accept" recommendation.

Reviewer #1: All comments have been addressed

Reviewer #2: All comments have been addressed

2. Is the manuscript technically sound, and do the data support the conclusions?

Reviewer #1: Yes

Reviewer #2: Yes

3. Has the statistical analysis been performed appropriately and rigorously?

Reviewer #1: Yes

Reviewer #2: Yes

4. Have the authors made all data underlying the findings in their manuscript fully available?

Reviewer #1: Yes

Reviewer #2: Yes

5. Is the manuscript presented in an intelligible fashion and written in standard English?

Reviewer #1: Yes

Reviewer #2: Yes

6. Review Comments to the Author

Reviewer #1: The authors have adequately addressed all the queries. I recommend that the editor accept the manuscript without any further revisions.

Reviewer #2: (No Response)

7. PLOS authors have the option to publish the peer review history of their article (what does this mean?). If published, this will include your full peer review and any attached files.

Reviewer #1: No

Reviewer #2: No

---

## [Editor Report · Acceptance letter]

PONE-D-26-09831R2

PLOS One

Dear Dr. Nguyen Bao,

I'm pleased to inform you that your manuscript has been deemed suitable for publication in PLOS One. Congratulations! Your manuscript is now being handed over to our production team.

Kind regards,

on behalf of

Dr. Anil Bhatia

Academic Editor

PLOS One